# Phenolic Compounds from *Mori Cortex* Ameliorate Sodium Oleate-Induced Epithelial–Mesenchymal Transition and Fibrosis in NRK-52e Cells through CD36

**DOI:** 10.3390/molecules26206133

**Published:** 2021-10-11

**Authors:** Yuan Ruan, Pei-Pei Yuan, Ya-Xin Wei, Qi Zhang, Li-Yuan Gao, Pan-Ying Li, Yi Chen, Yang Fu, Yan-Gang Cao, Xiao-Ke Zheng, Wei-Sheng Feng

**Affiliations:** 1College of Pharmacy, Henan University of Chinese Medicine, Zhengzhou 450046, China; ruanyuan8539@163.com (Y.R.); 15136153630@163.com (P.-P.Y.); 18439323730@163.com (Y.-X.W.); ZQfmaggily@163.com (Q.Z.); 15093153901@163.com (L.-Y.G.); lipanying2020@163.com (P.-Y.L.); chenyi9897@163.com (Y.C.); Fy18848890676@126.com (Y.F.); yxycyg@hactcm.edu.cn (Y.-G.C.); 2The Engineering and Technology Center for Chinese Medicine Development of Henan Province, Zhengzhou 450046, China

**Keywords:** *Mori Cortex*, CD36, diabetic nephropathy, NRK-52e cells, sodium oleate, EMT, fibrosis

## Abstract

Lipid deposition in the kidney can cause serious damage to the kidney, and there is an obvious epithelial–mesenchymal transition (EMT) and fibrosis in the late stage. To investigate the interventional effects and mechanisms of phenolic compounds from *Mori Cortex* on the EMT and fibrosis induced by sodium oleate-induced lipid deposition in renal tubular epithelial cells (NRK-52e cells), and the role played by CD36 in the adjustment process, NRK-52e cells induced by 200 μmol/L sodium oleate were given 10 μmoL/L moracin-P-2″-*O*-β-d-glucopyranoside (Y-1), moracin-P-3′-*O*-β-d-glucopyranoside (Y-2), moracin-P-3′-*O*-α-l-arabinopyranoside (Y-3), and moracin-P-3′-O-[β-glucopyranoside-(1→2)arabinopyranoside] (Y-4), and Oil Red O staining was used to detect lipid deposition. A Western blot was used to detect lipid deposition-related protein CD36, inflammation-related protein (p-NF-κB-P65, NF-κB-P65, IL-1β), oxidative stress-related protein (NOX1, Nrf2, Keap1), EMT-related proteins (CD31, α-SMA), and fibrosis-related proteins (TGF-β, ZEB1, Snail1). A qRT-PCR test detected inflammation, EMT, and fibrosis-related gene mRNA levels. The TNF-α levels were detected by ELISA, and the colorimetric method was used to detects SOD and MDA levels. The ROS was measured by flow cytometry. A high-content imaging analysis system was applied to observe EMT and fibrosis-related proteins. At the same time, the experiment silenced CD36 and compared the difference between before and after drug treatment, then used molecular docking technology to predict the potential binding site of the active compounds with CD36. The research results show that sodium oleate can induce lipid deposition, inflammation, oxidative stress, and fibrosis in NRK-52e cells. Y-1 and Y-2 could significantly ameliorate the damage caused by sodium oleate, and Y-2 had a better ameliorating effect, while there was no significant change in Y-3 or Y-4. The amelioration effect of Y-1 and Y-2 disappeared after silencing CD36. Molecular docking technology showed that the Y-1 and Y-2 had hydrogen bonds to CD36 and that, compared with Y-1, Y-2 requires less binding energy. In summary, moracin-P-2″-*O*-β-d-glucopyranoside and moracin-P-3′-*O*-β-d-glucopyranoside from *Mori Cortex* ameliorated lipid deposition, EMT, and fibrosis induced by sodium oleate in NRK-52e cells through CD36.

## 1. Introduction

Lipid deposition in the kidney can cause serious damage to the kidney—diabetic nephropathy is a classic example [1]. Diabetic nephropathy is one of the most important factors related to the deaths of diabetic patients [1]. Studies have shown that about 20% to 40% of diabetic patients will develop diabetic nephropathy [2,3], which is the second-leading cause of end-stage renal disease [4]. Diabetic nephropathy has become one of the most challenging problems in the world. One of the issues with treating diabetic nephropathy is that there are fewer drugs that target diabetic nephropathy alone, so the development of such drugs is essential.

Clinically, diabetic nephropathy is commonly detected by the appearance of lipid deposits in the kidneys [5]. Lipid deposition can induce cell lipotoxicity, further promote the occurrence of epithelial–mesenchymal transition (EMT) and fibrosis in the kidney, and seriously affect kidney function [6]. Sodium oleate is an unsaturated fatty acid that can induce lipid metabolism disorders and lipid deposition in HepG2 cells [7]. Studies have shown that sodium oleate has damage to the kidneys [8]. NRK-52e is the rat proximal tubular epithelial cell, which is often used to study kidney disease [9]. Therefore, this study used sodium oleate to stimulate NRK-52e cells to explore whether it could induce lipid deposition in NRK-52e cells.

In diabetic nephropathy, the kidney produces lipotoxicity due to lipid metabolism disorders, which will cause severe oxidative stress and inflammation [10]. As an important messenger for regulating lipid metabolism, CD36 plays a vital role in regulating the storage and use of fatty acids in the kidney [11]. Increased expression of CD36 will accelerate fatty acid intake, cause severe lipid metabolism disorders and accelerate kidney damage [12]. Most studies in recent years have shown that CD36 is an important target for lipid metabolism diseases [13].

The dry root bark of the dicotyledonous plant Morus alba L is called *Mori Cortex* after removing the cork [14]. It is cold in nature, sweet in taste [15]. There are records in ancient Chinese medicine books [15]. According to the literature, *Mori Cortex* has a hypoglycemic [16], lipid-lowering [17], anti-inflammatory [18], analgesic [19], and immune regulating [20] effect. Early in our laboratory, four new phenolic compounds with the same core were separated from the *Mori Cortex* water extract: moracin-P-2″-*O*-β-d-glucopyranoside (Y-1), moracin-P-3′-*O*-β-d-glucopyranoside (Y-2), moracin-P-3′-*O*-α-l-arabinopyranoside (Y-3), and moracin-P-3′-*O*-[β-glucopyranoside-(1→2)arabinopyranoside] (Y-4). Their structures are shown in Figure 1 [21]. Based on the hypoglycemic and lipid-lowering effects of *Mori Cortex*, this study investigated the intervention of the four compounds on NRK-52e cells induced by sodium oleate and then compared the effect difference with/without CD36 siRNA in the levels of oxidative stress, inflammation, EMT, and fibrosis. Specific reports will follow on whether the monomer compounds had an ameliorating effect and explore the possible potential targets and mechanisms.

## 2. Results

### 2.1. The Structure of the Drug

Y-1, Y-2, Y-3, and Y-4 are all phenolic compounds and with the same nucleus of moracin. The difference between Y-1 and Y-2 lies in the connection site of glucopyranosid, while Y-3 is connected to arabinopyranoside, and Y-4 is connected to a glucopyranosid more so than Y-3. The specific structure is shown in Figure 1.

### 2.2. Effects of Phenolic Compounds from Mori Cortex on Lipid Deposition of NRK-52e Cells Induced by Sodium Oleate

The NRK-52e cells in the normal control (NC) group were all polygonal, with clear edges and abundant cytoplasm. Oil Red O is a fat-soluble dye, which can highly dissolve lipids and stains specifically. CD36 is a fatty acid receptor protein responsible for fatty acid recognition and cell transmembrane transport and can reflect the level of lipid deposition. We used Oil Red O staining and detection of CD36 expression to characterize the effect of sodium oleate on NRK-52e lipid deposition. We did not observe the deposition of red oil droplets on the edges of the cells in the NC group. After stimulation with sodium oleate, the Oil Red O absorbance and the relative expression of CD36 protein in the model (M) group were significantly increased compared with the NC group (*p* < 0.01). Those given Y-1, Y-2, Y-3, and Y-4 interventions were named Y-1 group, Y-2 group, Y-3 group, and Y-4 group, respectively. Compared with the M group, the Oil Red O absorbance and the relative expression of CD36 protein in the Y-1 and Y-2 groups decreased (*p* < 0.05 or *p* < 0.01), and there was no significant change in the Y-3 and Y-4 groups, as shown in Figure 2. At the same time, we also tested the effect of sodium oleate stimulation on cell survival rate. The results showed that sodium oleate has no significant effect on NRK-52e cells survival rate, as shown in Appendix A. It is suggested that sodium oleate induced lipid deposition in NRK-52e cells and that Y-1 and Y-2 could alleviate the lipid deposition of the NRK-52e cells induced by sodium oleate.

### 2.3. Effects of Phenolic Compounds from Mori Cortex on Epithelial–Mesenchymal Transformation Induced by Sodium Oleate in NRK-52e Cells

The process of epithelial transformation to mesenchymal fibrocytes is called EMT, and its further development will cause the occurrence of organ fibrosis. CD31 protein is a marker of endothelial cells, and α-SMA is a marker of myofibroblasts. The disappearance of CD31 and the appearance of α-SMA are signs of the occurrence of EMT. We detected CD31 and α-SMA protein content and found that compared with the normal group, the expression of CD31 protein in the M group was extremely significantly downregulated (*p* < 0.01), and the expression of α-SMA protein was extremely significantly upregulated (*p* < 0.01). Compared with the M group, the expression of CD31 protein in Y-1 and Y-2 group were significantly increased (*p* < 0.05 or *p* < 0.01), and α-SMA expression was extremely significantly reduced (*p* < 0.01), and there was no significant change in the Y-3 and Y-4 groups, as shown in Figure 3A. It is suggested that sodium oleate induces EMT in NRK-52e cells. Y-1 and Y-2 can mitigate the EMT of NRK-52e cells induced by sodium oleate.

### 2.4. Effects of Phenolic Compounds from Mori Cortex on Fibrosis of NRK-52e Cells Induced by Sodium Oleate

Fibrosis can cause an increase in fibrous connective tissue in an organ or tissue and a decrease in parenchymal cells. Continued progress can lead to structural damage and functional decline of organs, and even failure. TGF-β is a key pro-fibrotic mediator in kidney disease. Zinc finger E box binding homeobox 1 (ZEB1) is a transcription factor located downstream of the TGF-β1 signaling pathway that regulates renal fibrosis. The transcription factor Snail 1, which is also located downstream of the TGF-β signaling pathway, is a DNA-binding protein containing a zinc finger structure. Its expression can accelerate the process of promoting EMT, thereby promoting renal fibrosis. We selected these three indicators to detect the fibrosis of our NRK-52e cells. The results showed that compared with the NC group, the expression of fibrosis-related proteins TGF-β ZEB1 and Snail1 in the M group was extremely significantly raised (*p* < 0.01). Compared with the M group, the ZEB1 TGF-β and Snail1 protein expression in the Y-1 and Y-2 group was significantly reduced (*p* < 0.05 or *p* < 0.01), and both the Y-3 and Y-4 groups did not obviously change, as shown in Figure 3B. It is suggested that sodium oleate induces fibrosis in NRK-52e cells. Y-1 and Y-2 can alleviate the fibrosis of NRK-52e cells induced by sodium oleate.

### 2.5. Effects of Y-1 and Y-2 on the Inflammatory Response of NRK-52e Cells Induced by Sodium Oleate after CD36 Silence

We have confirmed the inhibition of siCD36 on CD36 mRNA and protein levels in previous experiments, as shown in Appendix A. Based on this, we continued to design experiments to verify the role of CD36.

Inflammation can cause fever, and severe cases can cause acute multi-organ failure or death. Phosphorylation of NF-κB-P65 can activate the release of various pro-inflammatory factors of IL-1β and TNF-α in the body, causing severe inflammation. While testing the above indicators, we added the corresponding CD36 silent groups of each group for comparison. The results showed that compared with the NC group, the relative expression of NF-κB-P65, p-NF-κB-P65, and IL-1β protein and the TNF-α levels in the M group were significantly increased (*p* < 0.05 or *p* < 0.01). Compared with the M group, Y-1 and Y-2 could significantly reduce the above index changes (*p* < 0.01). Compared with M-siCD36, there were no significant changes in the Y-1-siCD36 and Y-2-siCD36 groups, as shown in Figure 4A,C. The qRT -PCR detection of NF-κB-P65 and IL-1β mRNA levels’ change trends was consistent with the WB results, as shown in Figure 4B. This indicated that sodium oleate-induced NRK-52e cells have an inflammatory response, and Y-1 and Y-2 may play a role in ameliorating that response through CD36 targets.

### 2.6. Effects of Y-1 and Y-2 on Oxidative Stress of NRK-52e Cells Induced by Sodium Oleate after CD36 Silence

Excessive oxidative stress can promote the development of fibrosis. The occurrence of oxidative stress is related to the imbalance of the antioxidant system, as well as the production of ROS and MDA. Nrf-2, Keap1, and SOD are important components of the antioxidant system in the body, and NOX1 can positively regulate the production of ROS. To explore whether our drugs alleviate oxidative stress through CD36, we silenced CD36 on NRK-52e and detected changes in oxidative stress. The results showed that compared with the NC group, the relative expression of NOX1 and Keap1 proteins and the MDA and ROS levels in the M group were significantly increased (*p* < 0.01), and the relative expression of Nrf2 protein, SOD levels were extremely significantly reduced (*p* < 0.01). Compared with the M group, Y-1 and Y-2 could significantly adjust the relative expressions of the Keap1 NOX1 and Nrf2 proteins and the levels of MDA SOD and ROS (*p* < 0.01 or *p* < 0.05). Compared with M-siCD36, the amelioration of Y-1 and Y-2 disappeared in the Y-1-siCD36 and Y-2-siCD36 groups, as shown in Figure 5. This suggests that sodium oleate-induced NRK-52e cells appear when there is oxidative stress and that Y-1 and Y-2 may play a role in ameliorating oxidative stress through CD36 targets.

### 2.7. Effects of Y-1 and Y-2 on Lipid Deposition and Epithelial–Mesenchymal Transition of NRK-52 Cells Induced by Sodium Oleate after CD36 Silencing

In order to further explore the target of drug action, we simultaneously detected the alleviating effect of drugs on the lipid deposition and epithelial–mesenchymal transition before and after silencing CD36. The results showed that compared with the NC group, the relative expression of CD31 protein in the M group was extremely significantly reduced (*p* < 0.01), and the Oil Red O absorbance and α-SMA protein was extremely significantly increased (*p* < 0.01). Compared with the M group, Y-1 and Y-2 could significantly adjust the relative expressions of CD31, α-SMA proteins, and the Oil Red O absorbance (*p* < 0.05 or *p* < 0.01). Compared with the M-siCD36 group, the Y-1-siCD36 and Y-2-siCD36 groups demonstrated no significant changes, as shown in Figure 6A,B. The change trend of CD31 and α-SMA mRNA levels detected by qRT-PCR were consistent with the results of WB, as shown in Figure 6C. The observation results of the high-content imaging analysis system were consistent with the above results, as shown in Figure 7. It is suggested that sodium oleate induces lipid deposition and EMT in NRK-52e cells and that Y-1 and Y-2 may play a role in ameliorating lipid deposition and EMT through the CD36 target.

### 2.8. Effect of Y-1 and Y-2 on Fibrosis of NRK-52 Cells Induced by Sodium Oleate after CD36 Silence

After silencing CD36, we tested fibrosis again; the results showed that the relative expression of TGF-β ZEB1 and Snail1 proteins in the M group were higher than those in the NC group (*p* < 0.01). Compared with the M group, Y-1 and Y-2 could significantly downregulate the relative expression of ZEB1 TGF-β and Snail1 proteins (*p* < 0.05 or *p* < 0.01). Compared with the M-siCD36 group, there was no significant change in the Y-1-siCD36 or Y-2-siCD36 groups, as shown in Figure 8A,B. The change trends of TGF-β ZEB1 and Snail1 mRNA levels detected by qRT-PCR were consistent with the WB results, as shown in Figure 8C. The observation results of the high-content imaging analysis system were the same as the above results, as shown in Figure 9. This suggests that sodium oleate-induced NRK-52e cells have fibrosis and that Y-1 and Y-2 may play a role in ameliorating fibrosis through the CD36 target.

### 2.9. Docking with CD36 Molecules

Molecular docking analysis shows that Y-1, Y-2, Y-3, and Y-4 have hydrogen bonds with CD36; the docking structure and interaction mode are shown in Figure 10. The lowest binding energy of Y-1 was −3.7 kcal/mol, Y-2 was −4.44 kcal/mol, Y-3 was −1.91 kcal/mol, and Y-4 was −1.41 kcal/mol. The binding force of Y-2 to CD36 was the strongest when compared with Y-1, Y-3, and Y-4.

## 3. Discussion

Diabetic nephropathy is accompanied by the appearance of lipid deposits. NRK-52e cells, in high glucose concentrations, will cause the initiation of epithelial–mesenchymal transition due to lipid deposition [22]. EMT is a key and basic biological behavior of cells, and it is also a key mechanism of renal interstitial fibrosis [23,24]. When the cells or organs undergo fibrosis, the connective tissues increase, and parenchymal cells are reduced. In severe cases, the structure of cells or organs may be damaged, with functional decline or even failure. In this study, cells without sodium oleate stimulation had no red oil droplets at their edges, and the expression of lipid metabolism-related protein CD36 was very low. After stimulating NRK-52e cells with sodium oleate, the Oil Red O staining and detection of lipid metabolism-related proteins showed that sodium oleate could induce lipid deposition in NRK-52e cells. Cells could accelerate the uptake of lipids in an environment of saturated or unsaturated fatty acids, causing disorders of lipid metabolism [25]. Therefore, does sodium oleate also cause high-risk biological behaviors of EMT and fibrosis in NRK-52e cells due to lipid metabolism disorders?

When renal cells start EMT due to external stimulation, the relative expression of mesenchymal marker α-SMA will increase, and the relative expression of endothelial phenotype CD31 of endothelial cells will be reduced, while the characteristics of myofibroblasts will occur [26]. Snail1 is a fibrosis marker in diabetic nephropathy and a key mechanism of EMT activation [27]. ZEB1 is an important transcription repressor that can induce EMT and fibrosis by inhibiting the expression of a variety of epithelial genes [28]. TGF-β1 is the main regulator of renal fibrosis [29]. Based on the above, this study used sodium oleate to induce NRK-52e cells to detect the levels of EMT and fibrosis-related proteins and found that the expression of α-SMA, Snail1, ZEB1, and TGF-β1 were extremely increased, and the expression of CD31 was extremely reduced, indicating that NRK-52e cells appear in EMT and fibrosis similar to diabetic nephropathy. This indicates that this study has successfully established an in vitro model of lipid deposition. In this study, Y-1 and Y-2 all ameliorated the lipid deposition, EMT, and fibrosis level in NRK-52e cells induced by sodium oleate.

It was found that the advanced glycation end-products of diabetic nephropathy stimulate the activation of a variety of signals, such as to the NF-κB family, which will further activate target genes: pro-inflammatory cytokines interleukin (IL-1β) and tumor necrosis factor-α (TNF-α) [30], which induce severe inflammation. At the same time, excessive peroxide and ROS exist in the kidneys of those with diabetic nephropathy [31], and NOX1 can derive ROS to positively regulate the activity of protein kinase C (PKC) subtypes in kidney tissue, while the activation of PKC signal will activate downstream NADPH. Oxidase, NADPH oxidase, is an important source of ROS [32,33]. Excessive ROS can cause damage to the kidney’s antioxidant system, Nrf-2-Keap1, decrease the activity of SOD, and increase MDA [34,35]. Can Y-1 and Y-2 improve the damage of sodium oleate to NRK-52e cells by alleviating inflammation and oxidative stress? In 2.6 to 2.8, we studied in detail the mechanism of our drug to improve the epithelial–mesenchymal transition and fibrosis of NRK-52e. Our results show that Y-1 and Y-2 can improve inflammatory response and oxidative stress. In terms of inflammation, we detected the activation of phosphorylated NF-κB-P65. The activation of phosphorylated NF-κB-P65 can activate NF-κB signaling [36]. We also detected high levels of downstream inflammatory cytokines IL-1β and TNF-α. Y-1 and Y-2 play an ameliorative role by inhibiting the phosphorylation of NF-κB-P65. At the same time, we also observed the high expression of NF-κB-P65 in NRK-52e induced by sodium oleate. It has been reported in the literature that high expression of NF-κB-P65 can cause damage to the kidney [37,38] and is correlated with the activation of the TGF-β pathway [39,40]. In our experiment, TGF-β expression was positively correlated with NF-κB-P65, and we verified this. In terms of oxidative stress, we first made it clear that the Nrf-2-Keap1 antioxidant system of NRK-52e was damaged after sodium oleate stimulation, and SOD activity was greatly reduced. Moreover, a large number of proteins that can mediate the production of ROS, such as NOX1, were highly expressive, and, at the same time, the level of ROS was significantly increased. Our drugs can significantly restore the antioxidant system and clear the source of ROS. This suggests that Y-1 and Y-2 may ameliorate the EMT and fibrosis of NRK-52e cells by alleviating inflammation and oxidative stress.

As a protein that promotes the uptake of long-chain fatty acids, fatty acid translocase CD36 has a key ability to regulate the homeostasis of fatty acids in cells [41,42]. Increasing CD36 expression in the results of this study suggested that NRK-52e cells could accelerate the uptake of fatty acids [43]. Y-1 and Y-2 are the compounds extracted from traditional Chinese medicine to reduce blood sugar and lipid levels. Do they directly or indirectly exert their ameliorating effect through CD36, the key protein of lipid metabolism? CD36 silencing of NRK-52e cells showed that the amelioration of the compounds disappeared for inflammation, oxidative stress, EMT, and fibrosis induced by sodium oleate in NRK-52e cells, indicating that Y-1 and Y-2 could be metabolized by lipids. Thus, CD36 was the key target to further improve inflammation and oxidative stress. From the structure of our four compounds, they have the same core: moracin. The difference lies in the type and position of the substituted sugars, as shown in Figure 1. Our test results show that Y-2 has the best amelioration effect, followed by Y-1, and Y-3 and Y-4 have no amelioration effect. We speculate that glucose and the core of moracin are the necessary structures for ameliorating EMT and fibrosis, and Y-3 and Y-4 are not active because they do not have the above-mentioned necessary structures. Different substitution methods may affect the stability of the compound. The stability of the drug is the key to its efficacy. The efficacy of our Y-1 and Y-2 proves this point. In Y-1, there is no conjugated system between glucose and the 2” hydroxyl group, and the reason for the better amelioration of Y-2 may be that glucose and the 3’hydroxyl group of the benzene ring form a P-π conjugated system. The structure is more stable. To verify the interaction between Y-1, Y-2, and CD36, further molecular docking was carried out. Molecular docking is a theoretical simulation method in the field of computer-aided drug research, which is often used to study the interaction ability and binding sites between molecules [44]. The research results show that Y-2 had the ability to interact with CD36 and binding sites. The binding force of Y-2 to CD36 was the strongest in our drug, which confirmed our hypothesis on the structure-function relationship.

In summary, sodium oleate can induce lipid deposition similar to diabetic nephropathy in NRK-52e cells, resulting in severe oxidative stress and inflammation, causing typical EMT and fibrosis. The phenolic compounds from *Mori Cortex*, moracin-P-2″-O-β-D-glucopyranoside (Y-1) and moracin-P-3′-O-β-D-glucopyranoside (Y-2), ameliorated the lipid deposition, EMT, and fibrosis of NRK-52e cells induced by sodium oleate through CD36. This discovery provides a new method for establishing an in vitro model of diabetic nephropathy and expands the basis of medication for diabetic nephropathy.

## 4. Materials and Methods

### 4.1. Materials

#### 4.1.1. Drugs

Y-1, Y-2, Y-3, and Y-4 were obtained from the aqueous extract system of *Mori Cortex* in the early stage of the laboratory [21]. The structure is shown in Figure 1.

#### 4.1.2. Reagents

DMEM (1898960, Gibco, Amarillo, TX, USA); SDS-PAGE gel preparation kit (64136301, Bio-Rad, Hercules, CA, USA); Lipo-2000 (1952313, Invitrogen, Waltham, MA, USA); Anti-CD31 (ab222783, Abcam, Cambridge, UK); Cy3 Goat Anti-Rabbit IgG (H + L) (AS007, Abclonal, Wuhan, China); Anti-CD36 (A1470, Abclonal, Wuhan, China); Anti-α-SMA (A17910, Abclonal, Wuhan, China); Anti-Snail1 (A5243, Abclonal, Wuhan, China); GAPDH (A19056, Abclonal, Wuhan, China); Anti-TGF-β (21898-1-AP, Proteintech, Wuhan, China); Anti-ZEB1 (21544-1-AP, Proteintech, Wuhan, China); Anti-NF-κB-P65 (AF5006, Affinify, Jiangsu, China); Anti-p-NF-κB-P65 (AF2006, Affinify, Jiangsu, China); Anti-IL-1β (AF5103, Affinify, Jiangsu, China); Anti-NOX1 (DF8684, Affinify, Jiangsu, China); Anti-Nrf2 (AF0639, Affinify, Jiangsu, China); Anti-Keap1 (AF5266, Affinify, Jiangsu, China); Normal rabbit serum for blocking (SL034, Beyotime, Shanghai, China); Cytoplasmic Membrane Extraction Kit (P0028, Beyotime, Shanghai, China); BeyoRT^TM^ III First Strand cDNA Synthesis Kit (D7178M, Beyotime, Shanghai, China); BCA Protein Quantification Kit (PC0020, Solarbio, Beijing, China); Total RNA Extraction Kit (R1200, Solarbio, Beijing, China); Reactive Oxygen Detection Kit (DCFH-DA) (CA1410, Solarbio, Beijing, China); DAPI solution (ready to use) (C0065, Solarbio, Beijing, China); Rat Tumor Necrosis Factor α Enzyme-Linked Immunoassay Kit (E-ELER2856c, Elabscience, Wuhan, China); Malondialdehyde Test Kit (A003-1, Nanjing Jiancheng Company, Nanjing, China); Superoxide Dismutase Assay Kit (A001-3, Nanjing Jiancheng Company, Nanjing, China); ChamQ Universal SYBR qPCR Master Mix (L/N8E271E8, Vazyme, Nanjing, China); OPTI-MEM (19571512, CellWorld, Beijing, China); sodium oleate (C10078860, Shanghai Maclean Biochemical, Shanghai, China); Oil Red O powder (O0625, Beijing Boao Tuoda Technology, Beijing, China); fetal bovine serum (Zhejiang Tianhang Biotechnology, Hangzhou, China); other chemical reagents are all commercially available analytical grades.

#### 4.1.3. Instruments

CO_2_ incubator (Thermo, Waltham, MA, USA); microplate reader (Bio-Tek, Winusky, VT, USA); real-time fluorescence quantitative PCR instrument (Applied Biosystems, Waltham, MA, USA); Odyssey dual-color infrared fluorescence imaging system (Odyssey, Nebraska, NE, USA); Mini-PROTEANTetra Electrophoresis tank (Bio-Rad, California, CA, USA); TransBlot Turb System all-round protein transfer instrument (Bio-Rad, California, CA, USA); high-content imaging analysis system (PerkinElmer, Waltham, MA, USA); AB204N precision analytical balance (Metler Toledo, Stockholm, Sweden); FlowSight multi-dimensional panoramic flow cytometer (Merck, Darmstadt, Germany); −80 °C ultra-low temperature refrigerator (Haier, Qingdao, China); −20 °C low temperature refrigerator (Haier, Qingdao, China).

### 4.2. Method

#### 4.2.1. Cell Culture

NRK-52e cell line, purchased from Shanghai Cell Bank of Chinese Academy of Sciences, cultured in high-sugar DMEM containing 10% fetal bovine serum, cultured in a CO_2_ incubator with 5% CO_2_, 37 °C, and saturated humidity; the logarithmic growth phase cells were taken to passage and experiment.

#### 4.2.2. Sodium Oleate Stock Solution and Oil Red O Dye Solution

Dissolved 50 mg of sodium oleate in 1800 μL of 0.1 mmoL/L NaOH solution. After fully dissolving, 5% BSA was added to make the total volume 10.8 mL, then filtered with a 0.22 μm membrane to obtain 15 mmoL/L sodium oleate stock solution [45].

A total of 75 mg of Oil Red O was added to 15 mL of 60% isopropanol and filtered with filter paper to obtain Oil Red O stock solution. Oil Red O stock solution and double-distilled water were combined in a volume of 3:2, then filtered with filter paper to obtain Oil Red O dye solution [46].

#### 4.2.3. Oil Red O

The NC group was cultured in DMEM containing 10% fetal bovine serum, and the M group and the administration group were given a certain concentration of sodium oleate and drug intervention under the same culture conditions. The specific intervention concentration and grouping are as follows: NC group, M group (200 μmoL/L sodium oleate), Y-1 group (200 μmoL/L sodium oleate + 10 μmoL/L Y-1), Y-2 group (200 μmoL/L sodium oleate + 10 μmoL/L Y-2), Y-3 group (200 μmoL/L sodium oleate +10 μmoL/L Y-3), and Y-4 group (200 μmoL/L sodium oleate +10 μmoL/L Y-4). When the cells’ density reached about 70%, the intervention in modeling and drug administration was performed. After 24 h, the Oil Red O experiment was performed. The lipid deposition was observed and photographed under an inverted microscope. The absorbance value was detected by a microplate reader for statistical analysis of the data.

#### 4.2.4. CD36 Transfection Silence

Cells were divided into the NC group and NC-siCD36 group. When the cell density was about 40%, the complete medium without antibiotics was replaced. After 6 h, the silence group was silenced by CD36 transfection in strict accordance with the instructions, and OPTI-MEM was added to the NC group as a control. The next day, the complete medium was replaced to simulate the modeling and administration. After 24 h, the total RNA extraction kit was used to extract each group of RNA, the BeyoRTTM III First Strand cDNA Synthesis Kit was used to reverse transcribe the sample into cDNA, and the ChamQ Universal SYBR qPCR Master Mix was used on the machine detection of CD36 and GAPDH mRNA levels. The primer sequences are shown in Table 1.

#### 4.2.5. Western Blot

The cells were divided into NC group, M group, and the administration group with/without CD36 siRNA. When the cell density was about 40%, the complete medium without antibiotics was replaced. After 6 h, the silent group was transfected with CD36 in strict accordance with the instructions for silence, and the non-silent group was added with OPTI-MEM as a control. Modeling and drug delivery were performed the next day, and the method was the same as Section 4.2.3. After 24 h, the cytoplasmic membrane extraction kit was used to extract the cell cytoplasm and membrane proteins, and the BCA protein quantification kit was used for protein quantification. In total, 35 μg of cytoplasm or membrane protein was loaded per well for 10% SDS-PAGE gel electrophoresis before it was semi-dry transferred to PVDF membrane, blocked with 5% BSA, and the primary antibodies CD36, NF-κB-P65, IL-1β, NOX1, Nrf2, Keap1, CD31, α-SMA, TGF-β, ZEB1, Snail1, and GAPDH were added. After overnight in 4 °C, the secondary antibody was added and incubated at room temperature. After cleaning the membrane, the Odyssey dual-color infrared fluorescence imaging system was used for scanning, and Image Studio software for statistical data analysis.

#### 4.2.6. qRT-PCR

The experimental grouping, silencing, and model-building intervention procedures were the same as in Section 4.2.5. After 24 h, RNA was extracted from each group. The method is the same as in Section 4.2.4. NF-κB-P65, IL-1β, CD31, α-SMA, TGF-β, ZEB1, Snail1, and GAPDH mRNA levels were detected by fluorescence via quantitative PCR. The primer sequences are shown in Table 2.

#### 4.2.7. TNF-α, SOD and MDA

The experiment grouping, silencing, and modeling administration intervention steps were the same as those in Section 4.2.5. After 24 h, the cells were collected by trypsin-free digestion with EDTA, and the cells were resuspended by adding PMSF-containing PBS. The cells were disrupted by ultrasound, and the supernatant was used for TNF-α enzyme-linked. For the immune kit, MDA test kit, and SOD determination kit, the instructions were strictly followed to detect the levels of TNF-α, SOD, and MDA in each group.

#### 4.2.8. Determination of ROS Levels

The experiment grouping, silencing, and modeling administration intervention steps were the same as those in Section 4.2.5. After 24 h, the cells were collected by trypsin digestion without EDTA, and the reactive oxygen detection kit (DCFH-DA) was used per the kit’s instructions, incubated for 20 min with 10 μmoL/L DCFH-DA in an incubator at 37 °C. Then, the ROS level of each group was checked on the computer.

#### 4.2.9. High-Content Imaging Analysis System to Observe Epithelial–Mesenchymal Transition and Fibrosis

The experiment grouping, silencing, and modeling administration intervention procedures were the same as those in Section 4.2.5. After 24 h, the cells were fixed with 4% paraformaldehyde at room temperature and blocked with normal rabbit serum and the primary antibodies CD31, α-SMA, TGF-β, ZEB1, and Snail overnight at 4 °C. The next day, Cy3 Goat Anti-Rabbit IgG and DAPI solution were added to fluorescently stain the cells, wash the excess dye in the wells and use the high-content imaging analysis system to scan and take pictures. Harmony 4.8 software was used to process and analyze the related proteins.

#### 4.2.10. Molecular Docking

Chemdraw software was used to make the 3D structure of the phenolic compounds in *Mori Cortex*, download the appropriate CD36 protein target (PDB ID: 5LGD) from the RCSB PDB database, and PyMOL process the original PDB protein molecule. AutoDock Tools software was used for molecular docking, selecting the most suitable conformation, and using PyMOL to make the 3D binding mode map.

### 4.3. Statistical Data Analysis

The data were statistically and analyzed using SPSS 20.0 software. The quantitative data were presented using single-factor ANOVA followed by *t*-test. The data were expressed as mean ± standard deviation ($\bar{x}$ ± s). One-way analysis of variance was performed between groups. *p* < 0.05 indicated statistically significant differences between the groups, and *p* < 0.01 indicated extremely significant differences.

## 5. Conclusions

Moracin-P-2″-*O*-β-d-glucopyranoside and moracin-P-3′-*O*-β-d-glucopyranoside from *Mori Cortex* ameliorated sodium oleate and induced lipid deposition, EMT, and fibrosis in NRK-52e cells through CD36.

## Figures and Tables

**Figure 1 molecules-26-06133-f001:**
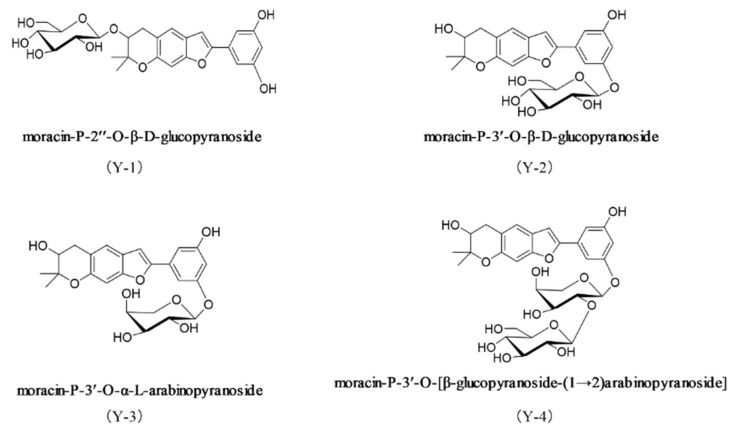
The structures of four phenol compounds isolated from the water extract of *Mori Cortex*.

**Figure 2 molecules-26-06133-f002:**
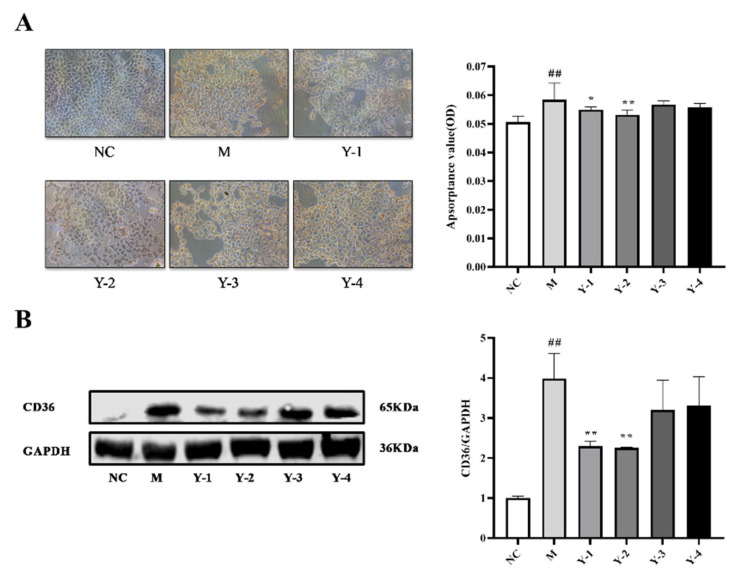
Effects of phenolic compounds from *Mori Cortex* on NRK-52e cells’ lipid deposition stimulated by sodium oleate. Effect of sodium oleate on lipid accumulation in NRK-52e cells detected by Oil Red O staining (**A**). The expression of CD36 protein in NRK-52e cells were quantified by Western blot and normalized (*n* = 3) (**B**); ^##^
*p* < 0.01 compared with NC group, * *p* < 0.05, ** *p* < 0.01 compared with M group.

**Figure 3 molecules-26-06133-f003:**
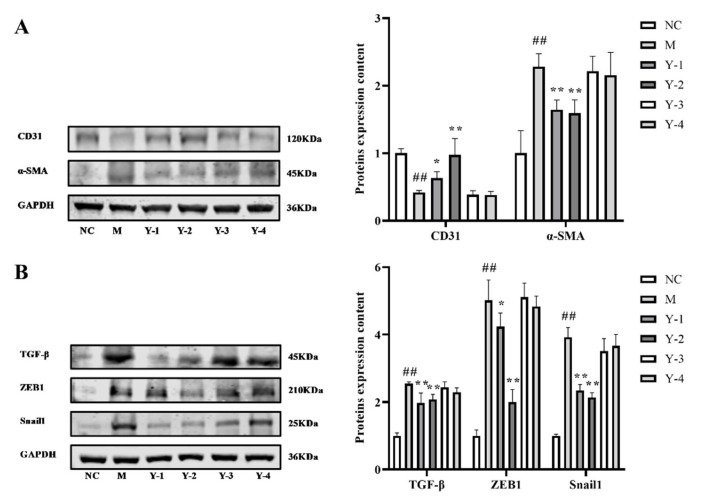
Effects of phenolic compounds from *Mori Cortex* on NRK-52e cells’ EMT and fibrosis stimulated by sodium oleate. The expression of CD31 and α-SMA protein in NRK-52e cells were quantified by Western blot and normalized (*n* = 3) (**A**). The expression of TGF-β ZEB1 and Snail1 protein in NRK-52e cells were quantified by Western blot and normalized (*n* = 3) (**B**); ^##^
*p* < 0.01 compared with NC group, * *p* < 0.05, ** *p* < 0.01 compared with M group.

**Figure 4 molecules-26-06133-f004:**
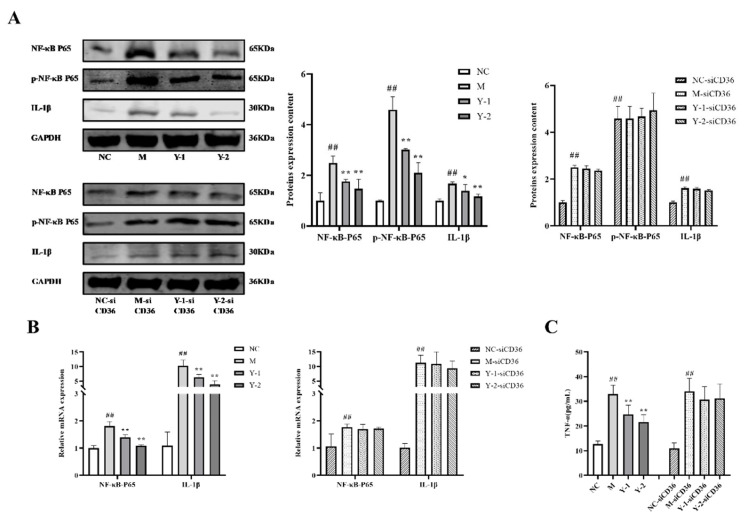
Effects of Y-1 and Y-2 on NRK-52e cells’ inflammatory cytokines stimulated by sodium oleate with/without CD36 silences representative. The expression of NF-κB-P65, p-NF-κB-P65, and IL-1β protein in NRK-52e cells were quantified by Western blot and normalized (*n* = 3) (**A**). The expression of NF-κB-P65 and IL-1β mRNA in NRK-52e cells were quantified by qRT-PCR and normalized (*n* = 3) (**B**). The levels of TNF-α in NRK-52e cells were determined by ELISA method (*n* = 6) (**C**). Compared with NC or NC-sicD36 group, ^##^
*p* < 0.01; compared with M or M-sicD36 group, * *p* < 0.05, ** *p* < 0.01.

**Figure 5 molecules-26-06133-f005:**
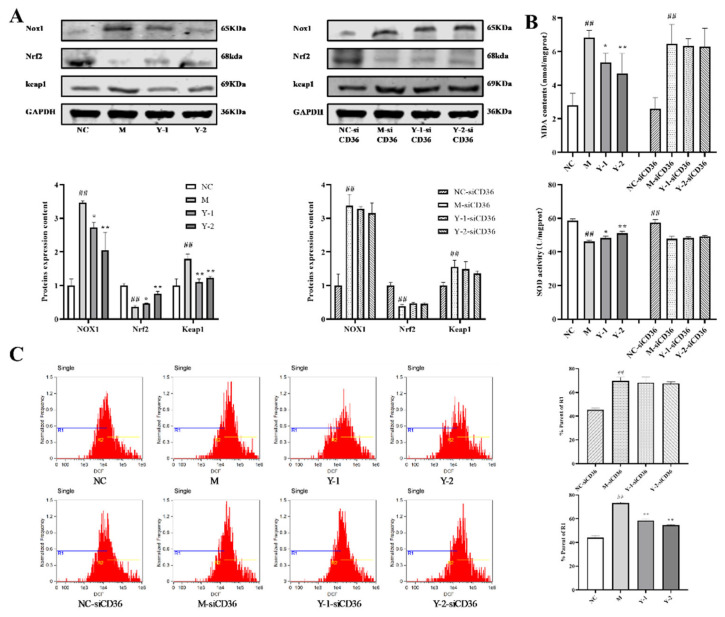
Effects of Y-1 and Y-2 on oxidative stress of NRK-52e cells stimulated by sodium oleate with/without CD36 silences. The expression of NOX1, Nrf2, and Keap1 protein in NRK-52e cells was quantified by Western blot and normalized (*n* = 3) (**A**). The levels of SOD and MDA in NRK-52e cells were determined by colorimetric method (*n* = 6) (**B**). The levels of ROS in NRK-52e cells were determined by FlowSight multi-dimensional panoramic flow cytometer (*n* = 3) (**C**). Compared with NC or NC-sicD36 group, ^##^
*p* < 0.01; compared with M or M-sicD36 group, * *p* < 0.05, ** *p* < 0.01.

**Figure 6 molecules-26-06133-f006:**
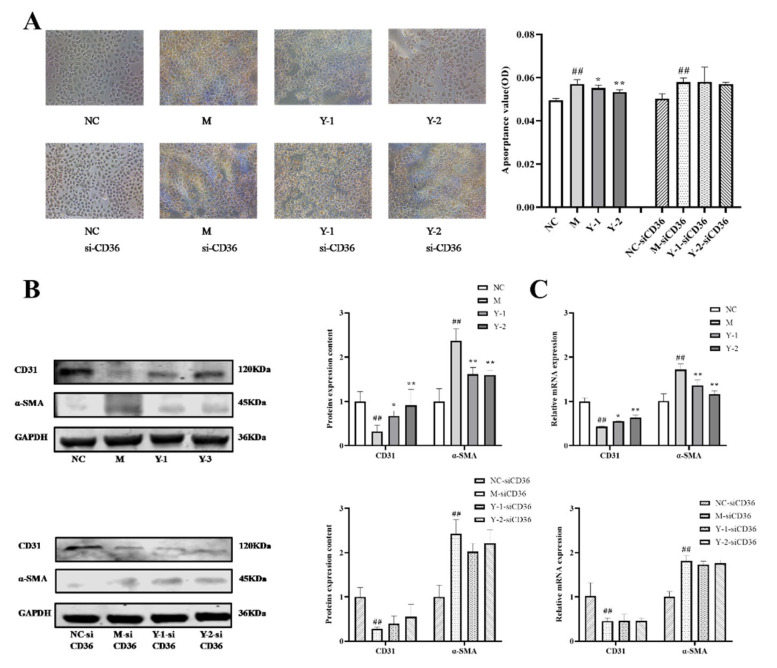
Effects of Y-1 and Y-2 on lipid deposition and EMT of NRK-52e cells stimulated by sodium oleate with/without CD36 silencing. Effect of sodium oleate on lipid accumulation in NRK-52e cells detected by Oil Red O staining (**A**). The expression of CD31 and α-SMA proteins in NRK-52e cells were quantified by Western blot and normalized (*n* = 3) (**B**). The expression of CD31 and α-SMA mRNA in NRK-52e cells were quantified by qRT-PCR and normalized (*n* = 3) (**C**). Compared with NC or NC-sicD36 group, ^##^
*p* < 0.01; compared with M or M-sicD36 group, * *p* < 0.05, ** *p* < 0.01.

**Figure 7 molecules-26-06133-f007:**
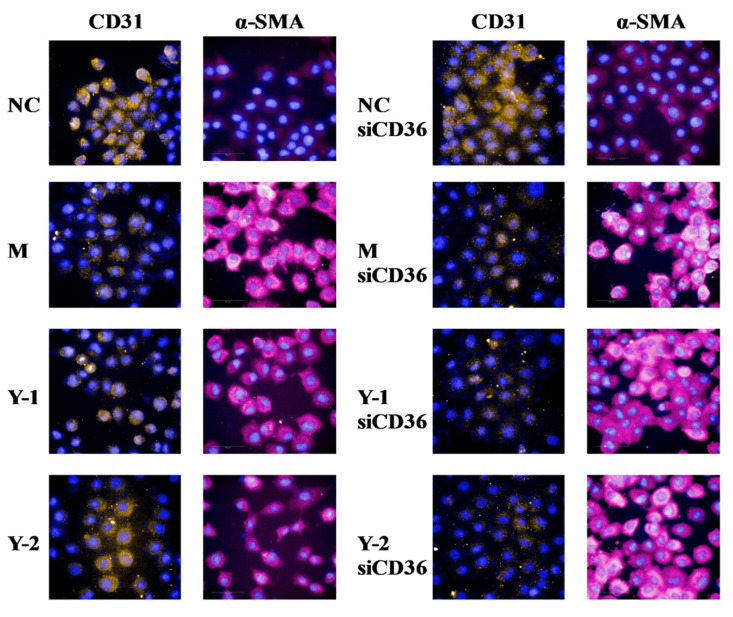
Cell immunofluorescence of EMT-related proteins CD31 and α-SMA.

**Figure 8 molecules-26-06133-f008:**
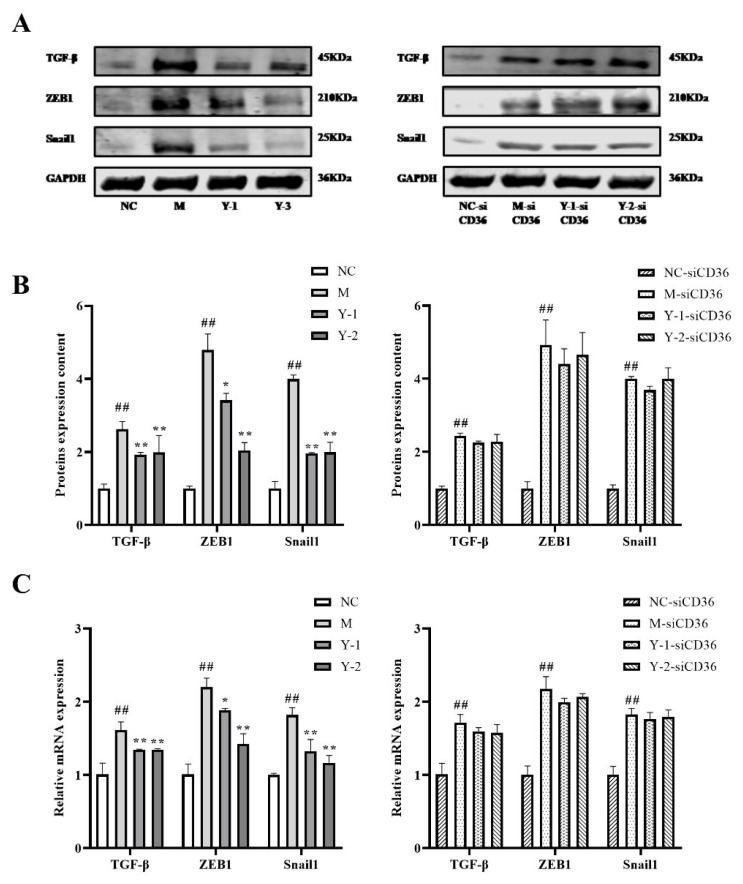
Effects of Y-1 and Y-2 on the fibrosis of NRK-52e cells stimulated by sodium oleate with/without CD36 silencing. The expression of TGF-β ZEB1 and Snail1 proteins in NRK-52e cells were quantified by Western blot and normalized (*n* = 3) (**A**,**B**). The expression of TGF-β ZEB1 and Snail1 mRNA in NRK-52e cells were quantified by qRT-PCR and normalized (*n* = 3) (**C**). Compared with NC or NC-sicD36 group, ^##^
*p* < 0.01; compared with M or M-sicD36 group, * *p* < 0.05, ** *p* < 0.01.

**Figure 9 molecules-26-06133-f009:**
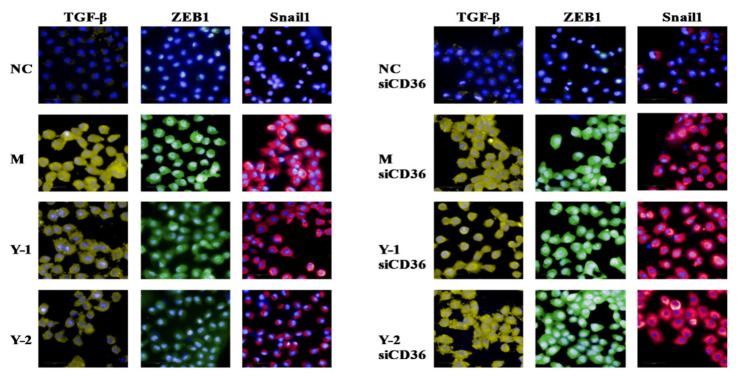
Cell immunofluorescence of fibrosis-associated proteins TGF-β, ZEB1, and Snail1.

**Figure 10 molecules-26-06133-f010:**
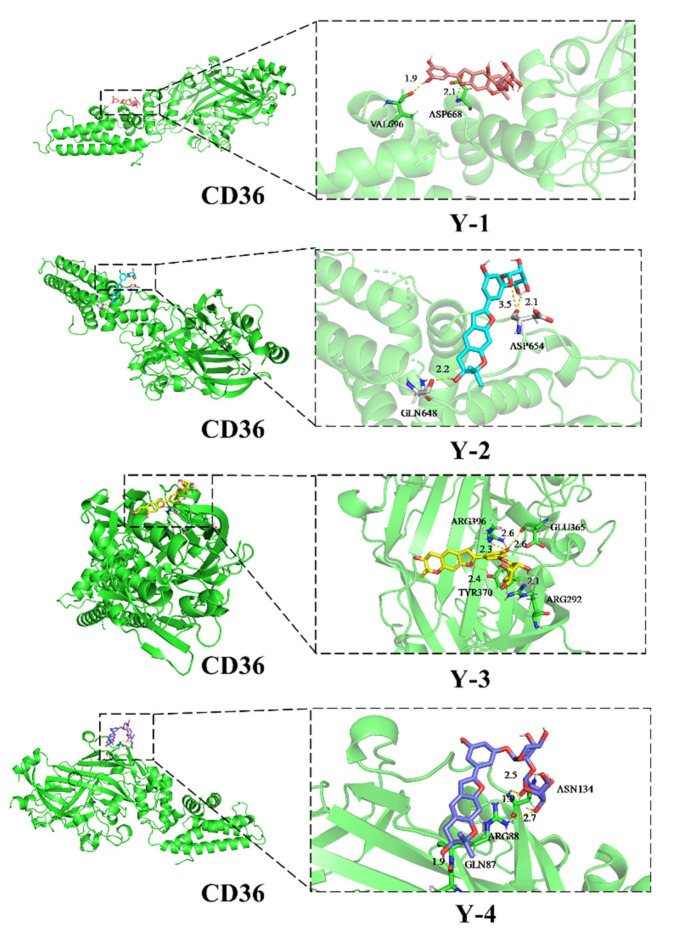
The binding mode of Y-1 and Y-2 with CD36 based on molecular simulation.

**Table 1 molecules-26-06133-t001:** List of primers and their sequences used in this study.

Gene	Forward Primer	Reverse Primer
CD36	AAGGCUCAAAGAUGGCUCCTT	AAGGCUCAAAGAUGGCUCCTT
GAPDH	ACAGCAACAGGGTGGTGGAC	TTTGAGGGTGCAGCGAACTT

**Table 2 molecules-26-06133-t002:** List of primers and their sequences used in this study.

Gene	Forward Primer	Reverse Primer
NF-κB-P65	TGTATTTCACGGGACCTGGC	CAGGCTAGGGTCAGCGTATG
IL-1β	AGGCTGACAGACCCCAAAAG	CTCCACGGGCAAGACATAGG
CD31	CAGCCATTACGACTCCCAGA	GAGCCTTCCGTTCTCTTGGT
α-SMA	ACCATCGGGAATGAACGCTT	CTGTCAGCAATGCCTGGGTA
TGF-β	GACTCTCCACCTGCAAGACC	GGACTGGCGAGCCTTAGTTT
ZEB1	GTGGATGGAAATGAGCCCCA	ACACAAGAGTAACCCTGCGG
Snail1	GAGGCCTTCATTGCCTTCCC	CCCAGGCTGAGGTACTCCTTA
GAPDH	ACAGCAACAGGGTGGTGGAC	TTTGAGGGTGCAGCGAACTT

## Data Availability

The date presented in this study are available on request from the corresponding author.

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
