# Peer review of "Phenolic Compounds from Mori Cortex Ameliorate Sodium Oleate-Induced Epithelial–Mesenchymal Transition and Fibrosis in NRK-52e Cells through CD36"

_molecules, 2021, doi:10.3390/molecules26206133_

Round 1

Reviewer 1 Report

The aim of the present study is not clear. Especially taking into account numerous experiments performed by Authors. Please clarify.

The manuscript requires extensive English proofreading. The text is very difficult to follow due to a poor English usage that leads to conceptual errors throughout the manuscript. 

Reviewer 2 Report

Potential study reports the treatment with phenolic compounds from Mori cortex in renal damage in vitro model. However, the grammar and sentence structure of the complete manuscript needs to be revised by a native English speaker.

The results demonstrated the effect of these compounds in reducing lipid deposition, fibrosis and inflammatory markers, and oxidative stress. The results suggest the participation of CD36 in the mechanism induced by these compounds, nonetheless in a non-conclusive way.

Some important points that must be reviewed:

Page 1/ line 27: “drug treatment”;

Page 2/ line 73: describe NC group;

Page 4: figure 3A definition;

Page 5/ line 115: subitem 2.4 put in the supplementary results;

Page 7: figure 7C and D impossible to visualize; figure E rename the axes;

Page 9/ line 177: adjust the subtitle;

Page 11/ line 197: adjust the subtitle;

Page 11/ line 198: introduce dock with Y3 and Y4;

Page 11/ line 205: adjust the subtitle;

Page 12/ line 213: remove (EMT);

Page 13/ line 278: add (figure 1);

Page 14/ line 323: describe NC and M;

Page 16/ line 393: describe data normality.

Reconsider in all the text the word “IMPROVE.”

Why the effect of Y3 and Y4 compounds were not demonstrated after CD36 silencing?

Discuss the structure-activity relationship for Y1/Y2 and Y3/Y4 to justify the group results.

Reconsider the affirmative sentence “CD36 mediated”.

Reviewer 3 Report

1) Although the authors claimed in the introduction that the effects of sodium oleate on the kidneys are rarely reported, some reports have mentioned the presence of toxicity in a rodent model, such as PMID 23272807.

2) The illustration of  Y-1 to Y-4 structures (Figure 1) is recommended being shown in the Results section.

3) What is M group? How is M treated? Not specified in the text and legend. What are the groups treated with sodium oleate or sodium oleate combined with Y-1-to-4? The descriptions are vague, rendering poor legibility for the readers.

4) Was cell viability examined under the experimental setting?

5) As phospho-NF-kB-p65 is required for the transcriptional activity to trigger inflammatory signaling, the concurrent measurement of its phosphorylated and total form is more robust.

6) As a pro-inflammatory cytokine, IL-1B primarily exerts it biological action after caspase-mediated maturation and extracellular secretion. Was IL-1B expression in this model determined by real-time PCR or ELISA?

7) Y1/Y2 act to decrease CD36. The ROS-suppressing effect of Y1/Y2 was abrogated by siCD36. That suggests the effect of Y1/Y2 on ROS requires CD36. It seems to me a paradoxical finding. The authors need to cautiously interpret the data.

8) The CD31 and a-SMA data are repeatedly shown in Figure 1 and Figure 8. The authors are advised to considerably reconstruct the manuscript to avoid redundant expression.

9) Overall this manuscript can be further constructed and improved. 

Reviewer 4 Report

Yuan Ruan and colleagues submitted the manuscript, "Phenolic compounds from Mori Cortex improve sodium oleate-induced epithelial-mesenchymal transition and fibrosis in NRK-52e cells by mediating CD36". The authors show that some phenolic compounds from Mori Cortex avoided the impact of sodium oleate in a rat kidney cell line. The results are solid, and the manuscript fits with the scope of Molecules. However, there are several issues to solve before accepting the work for publication. Notably, all the sections of the manuscript should be reformulated to increase the quality of the presentation. I include below the concerns that I consider that you need to address:

Major concerns (format):

  1. The introduction should be improved, linking the different paragraphs and justifying the aim of the work.
  2. The results need to be better explained. For example, the first time you mention NC or M group or the cell line should be defined (section 2.1). An introductory sentence is also hardly recommended at the beginning of each section.
  3. You should introduce the role of the different proteins (for example, CD31 in section 2.2)
  4. The figure legends are not detailed enough. It is missed the name of proteins detected in the western blots. It is not mentioned the loading control, and which is the relative value when the data is relativized. In figure legend 7, you should include how ROS levels were measured.
  5. The discussion is a summary of the results. I encourage you to improve the discussion with a deeper analysis of the main results. The first paragraph of the discussion fits better in the introduction.
  6. The number of figures is high, and some figures could be combined (Figure 3 and 4, Figure 5 and 6, Figure 8 and 9)

Major concerns (experimental):

  1. Since CD36 plays a role as fatty acid transporter, I suggest monitoring the CD36 levels at the surface. There are several suitable fluorescent antibodies to observe in live cells by FACS or microscopy.
  2. Which is the decrease of CD36 protein levels after silencing?
  3. In Figure 2, CD36 is not detected in NC group, but mRNA is detected according to Figure 5. This contradiction should be addressed or discussed in the manuscript.
  4. The reduction of fatty acid content in CD36-downregulated cells should be presented similar to Figure 2A to confirm the role of CD36 after sodium oleate treatment.

Minor concerns:

  1. Title: Improve without s.
  2. Line 166: Remove “High content”
  3. Line 236: “It is” and not “it’s”
  4. Section 4.1.2: The antibodies should be referred “anti- + name of the protein”, not only the protein
  5. I suggest improving to transfection silence section for better understanding.
  6. Section 4.2.8: How much time do you incubate DCFHDA and at which concentration?
  7. The quality of some westerns could be better.

Round 2

Reviewer 1 Report

I accept the manuscript in present form

Author Response

Dear reviewer,

Thank you for your suggestions and approval of this manuscript. Your suggestions are of great help to this manuscript, and of great guiding significance to our future experimental design.

If you have any question about this manuscript, please don't hesitate to contact with us.

Many thanks for your time and consideration.

Sincerely,

Wei-Sheng Feng

Tel: +86-371-6019-0296

School of Pharmacy, Henan University of Chinese Medicine.

No.156 Jinshui East Road, Zhengzhou, Henan Province, China

Reviewer 2 Report

Accept in present form

Author Response

(The authors gave the same response as above.)

Reviewer 3 Report

The authors have addressed all the issues and made substantial amendments. As such, I recommend acceptance for publication. 

Author Response

(The authors gave the same response as above.)

Reviewer 4 Report

Dear authors,

I’m very grateful for your detailed answers. I appreciate that all the concerns in format were solved. You already included new experiments in figure 6A. However, I consider that there is still a concern to solve. The validation of your siRNA is neat (now included in a supplementary figure), but I still think you should monitor the protein levels according to the low CD36 levels in NC cells. You may not have a significant reduction in NC cells, but yes in the other groups. As a minor concern, it is still indicated in Figure 5 in Supplementary Fig 1.  
